Non-lethal approach identifies variability of δ15N values in the fin rays of Atlantic Goliath Grouper, Epinephelus itajara

Tzadik Orian E. 1 otzadik@mail.usf.edu
Goddard Ethan A. 1
Hollander David J. 1
Koenig Christopher C. 2
Stallings Christopher D. 1
1 College of Marine Science, University of South Florida , St Petersburg, FL , USA
2 Coastal and Marine Laboratory, Florida State University , St Teresa, FL , USA
Esteban María Ángeles
Electronic publication date: 2015 Jun 11
Publication date: 2015
Volume: 3
Electronic Location ID: e1010
Received 2015 Apr 9; Accepted 2015 May 19
Copyright: © 2015 Tzadik et al.
Copyright year: 2015
Copyright holder: Tzadik et al.
License: This is an open access article distributed under the terms of the Creative Commons Attribution License, which permits unrestricted use, distribution, reproduction and adaptation in any medium and for any purpose provided that it is properly attributed. For attribution, the original author(s), title, publication source (PeerJ) and either DOI or URL of the article must be cited.
License URL: https://creativecommons.org/licenses/by/4.0/

Keywords: Isotope chronology, Fin-ray chemistry, Ontogeny, Trophic shifts, Nursery, Mangrove habitat, Food web, Diet

Funding: MARFIN NA11NMF4330123 Marine Resource Assessment Fellowship NA10NMF4550468 This study was supported by the U.S. National Oceanic and Atmospheric Administration through a MARFIN award to CC Koenig and CD Stallings (NA11NMF4330123) and a Marine Resource Assessment Fellowship to the senior author (NA10NMF4550468). The funders had no role in study design, data collection and analysis, decision to publish, or preparation of the manuscript.

==============================
The Atlantic Goliath Grouper, Epinephelus itajara, is critically endangered throughout its range but has begun to show initial signs of recovery in Florida state waters. As the population continues to rebound, researchers face a pressing need to fill the knowledge gaps about this iconic species. Here, we examined the δ15N isotopic records in fin rays collected from Atlantic Goliath Grouper, and related changes of isotopic ratios over time to life history characteristics. Fin-ray analysis was used as a non-lethal technique to sample individuals from two locations at similar latitudes from the west and east coasts of Florida, USA. δ15N data were acquired by mechanically separating the annuli of each fin ray and then analyzing the material in an Irradiance Elemental Analyzer Mass Spectrometer. The δ15N values were consistent among individuals within populations from each coast of Florida, and mirrored the expected changes over the lives of the fish. Overall, differences were found between δ15N values at juvenile life history phases versus adult phases, but the patterns associated with these differences were unique to each coastal group. We demonstrated, for the first time, that δ15N values from fin rays can be used to assess the life histories of Atlantic Goliath Grouper. The non-lethal strategies outlined here can be used to acquire information essential to the management of species of concern, such as those that are threatened or endangered.

Introduction

The Atlantic Goliath Grouper (Epinephelus itajara) is the largest epinephelid in the Atlantic Ocean and the second largest in the world, weighing up to 400 kg and reaching lengths of up to 3.0 m (Bullock et al., 1992; Robbins, Ray & Douglass, 1999; Sadovy & Eklund, 1999). They are a long-lived, slow growing fish that can remain in their juvenile habitat (primarily mangroves) for up to 7 years before moving to reef habitats as adults (Bullock et al., 1992; Koenig et al., 2007). Unlike other large reef fishes which tend to be upper trophic-level piscivores (Romanuk, Hayward & Hutchings, 2011), invertebrates make up approximately 70% of the diet of the Atlantic Goliath Grouper (Koenig & Coleman, 2010). These large fish can serve as ecological engineers by exposing and expanding reef overhangs and ledges through their excavating activities. This behavior enhances structural complexity of the habitat thereby increasing abundance and diversity of the reef community (Koenig, Coleman & Kingon, 2011; Macieira et al., 2010).

Similar to many large-bodied reef fishes which are vulnerable to overfishing (e.g., due to slow maturation, aggregation behavior, limited juvenile habitat; Stallings, 2009), Atlantic Goliath Grouper are overfished throughout their range (Aguilar-Perera et al., 2009; McClenachan, 2009) and are classified as “critically endangered” by the International Union for Conservation of Nature (Pusack & Graham, 2009). However, the population of Atlantic Goliath Grouper has shown early signs of recovery in Florida state waters, in large part due to a federal fishing moratorium instituted in the United States in 1990 (Cass-Calay & Schmidt, 2009; Koenig, Coleman & Kingon, 2011). While this initial recovery is encouraging, more basic research on life history traits is needed to enhance and inform management. However, the slow maturation, large size, behavior, and long lifespans of Atlantic Goliath Groupers limit our ability to infer processes from controlled experimentation and short observational studies. To date, movement patterns and trophic shifts from nursery to adult habitats are still poorly understood, and warrant further investigation (Lara et al., 2009; Koenig & Coleman, 2010). The study of these processes in fishes typically requires lethal sampling, however due to the endangered status of the Atlantic Goliath Grouper, a non-lethal sampling technique is needed.

Stable isotope analysis (SIA) has become a common method to study fish movements and diets. In fishes, muscle tissue is most commonly used to quantify basal resources (δ13C; Hobson, 1999; Dierking et al., 2012) and trophic level (δ15N; Vanderklift & Ponsard, 2003; Galvan, Sweeting & Reid, 2010). Because these isotopic ratios integrate chemical information about an animal’s diet across time scales beyond the “snapshot” scale from examining stomach contents, they can be used to quantify dietary patterns over a period of weeks to months (Nelson et al., 2011), before tissue turnover (Ankjaero, Christensen & Gronkjaer, 2012; Hobson & Bond, 2012). However, the study of long-lived fishes requires knowledge of longer time frames, often on the order of years.

To understand life history characteristics over annual time scales, researchers have recognized the need to analyze a conserved organic matrix that retains isotopic ratio values over the entire lifetime of the individual (Caut, Angulo & Courchamp, 2008). To our knowledge, a chronology of isotopic ratios continuously from birth to time of capture for an individual fish has only been accomplished via lethal sampling methods. Wallace, Hollander & Peebles (2014) measured both δ13C and δ15N across sequentially deposited layers of the eye lenses. However, this method has only recently been validated and the time scale over which eye-lens layers are deposited remains unclear. The sagittal otolith has been suggested as another possibility due to its chronological deposition of a metabolically inert matrix (Campana, 1999). However, the otolith contains miniscule amounts of organic material that may be conserved chronologically. To date, otoliths have only been analyzed at the bulk level for the entire structure, thus destroying the time series of interest (Gronkjaer et al., 2013). Otolith sampling also requires the fish to be sacrificed, thus confounding conservation and management efforts for threatened and endangered species. Cartilaginous vertebrae in elasmobranches (Estrada et al., 2006; Borrell et al., 2011; Polo-Silva et al., 2013) have been used to document life history characteristics, however sampling is also lethal, and annuli banding in elasmobranch vertebrae are often difficult to interpret or absent for many species (Cailliet et al., 2006). While scales in teleosts (Kennedy et al., 2005; Kelly et al., 2006; Sinnatamby, Dempson & Power, 2008; Woodcock & Walther, 2014) represent non-lethal sampling, they may present inaccurate age estimations as the annuli of older fish tend to compress at the edge of the scales (more than other calcified structures). Additionally, scales are often lost and replaced, and the formation of scales may not occur at the larval stage (Helfman et al., 2009).

Fin rays of fishes can record the chronology of isotopes and may allow for non-invasive sampling as they can be excised non-lethally. Indeed, fin rays have the capability to regrow once they are excised (Goss & Stagg, 1957) and can be removed with minimal effects on both survival and growth on the individual (Zymonas & McMahon, 2006). The organic matrix of fin rays is largely composed of proteins, mostly collagen, while the inorganic matrix is carbonated hydroxyapatite (Mahamid et al., 2010). Chemical tracers from an individual’s diet have been recorded over time within these matrices, suggesting that they are at least partially derived from the animal’s food source (Woodcock, Grieshaber & Walther, 2013). Annuli conservation over time and the encapsulation of the organic matrix suggest that isotopic values of organic elements (e.g., δ13C, δ15N) are retained within these matrices. Initially, aging studies concentrated on the analysis of fin rays for fishes in temperate regions, such as salmonids and hexagrammids, aided by clear banding of annuli due to strong seasonality (Bilton & Jenkinson, 1969; Beamish & Chilton, 1977). However, more recent studies have demonstrated the effectiveness of the technique on fishes at lower latitudes, including Gag and Goliath Groupers (Murie & Parkyn, 2005; Brusher & Schull, 2009; Murie et al., 2009; Koenig et al., 2011). The mineral deposition in fin rays occurs on a similar time scale as in otoliths, although the two structures are formed via different metabolic pathways (Helfman et al., 2009). Indeed, age estimates from cross sections of fin rays corresponded to those from otoliths of the same fish (McFarlane & King, 2001; Muir et al., 2008; Khan & Khan, 2009; Murie et al., 2009; Glass, Corkum & Mandrak, 2011). The correspondence of annuli between otoliths and fin rays suggests minimal turnover or reabsorption in fin rays since previous layers are encapsulated and non-vascularized after new ones are added. The fin ray comprises both organic and inorganic chemical matrices, with a robust organic component (∼40%) compared to other calcified structures (Mahamid, 2010).

The documentation of a conserved organic matrix over time via fin-ray analysis may provide essential information regarding ontogenetic dietary and movement patterns of fishes. The method is suited to study life history characteristics for endangered species, such as the Atlantic Goliath Grouper, and those of management concern, due to its non-lethal nature and conserved chemical history. We tested whether δ15N values were retained over time in fin rays of Atlantic Goliath Grouper, and if these changes were consistent with life history characteristics documented in previous studies. Changes in δ15N values over time within an individual can be caused by movement to areas with different isotopic baselines and dietary shifts. The documentation of these changes can be used to guide strategies that minimize the impact to the local population via habitat restoration and responsible fishing practices. Considering the lack of information on these ontogenetic characteristics of Atlantic Goliath Grouper, the technique presented in this study may lay the foundation for future research as well as inform management.

Materials and Methods

Study area

We obtained samples of fin rays from adult Atlantic Goliath Grouper from mid-Peninsular regions of Florida on both the Gulf of Mexico (hereafter, “west coast fish”) and Atlantic Ocean sides (hereafter, “east coast fish”). West coast fish (n = 13) were acquired from the Florida Fish and Wildlife Research Institute (FWRI) when opportunistic “fish-kill” samples (e.g., red tide casualties, discard mortalities) were reported from May 2012 to September 2013 (Site 1, Fig. 1). East coast fish (n = 17) were collected at known spawning aggregation sites during spawning seasons (July–September) in 2012 and 2013 (Site 2, Fig. 1). All samples were obtained through the procedure approved by the Institutional Animal Care and Use Committee (IACUC), approval number 4193W. In addition, all field sampling was permitted on both the state (Florida Fish and Wildlife Conservation Commission, permit number SAL-13-1244A-SRP) and federal levels (National Oceanic and Atmospheric Administration, permit number F/SER24:PH). Sites with elevated Atlantic Goliath Grouper abundances were chosen based on local knowledge and later confirmed by SCUBA surveys during the spawning season. Sites were typically artificial reefs (sunken wrecks) or natural ledges with high structural relief.

Figure 1 Sampling regions.

Regions on the west and east coasts of Florida where Atlantic Goliath Grouper (Epinephelus itajara) were sampled (gray); mangroves are shown in black (Florida Fish and Wildlife Research Institute, 2013).

Tissue selection and sample collection

The soft-dorsal fin rays of the Atlantic Goliath Grouper were chosen for analysis over other calcified structures for several reasons. First, fin rays will grow back once they are excised (Goss & Stagg, 1957). The effects of fin ray removals do not significantly alter survival or growth of individuals (Zymonas & McMahon, 2006). Indeed, during the current study, several individuals were recaptured the same day after having their fin rays excised (indicating feeding behavior within hours of being sampled) and we have recaptured several fish over 1,100 days after initial sampling (C Koenig et al., 2014, unpublished data). Our methodology for capturing, excising the fin rays and release of the Atlantic Goliath Grouper has resulted in approximately 100% survival (Koenig et al., 2011). When compared to other calcified structures such as fin spines and scales, fin rays have the highest correspondence to ages obtained from otoliths in Atlantic Goliath Groupers (Murie et al., 2009). In addition, the organic matrix in fin rays is proportionally larger than any other reliable chronological recorder in Atlantic Goliath Grouper and many other fishes. Last, fin rays were being used for aging in a collaborative study, and were already being excised for analysis (Koenig et al., 2011).

Excision of dorsal fin-rays was deemed preferable to other fins due to the relative low usage of this fin during locomotion in Atlantic Goliath Groupers. Dorsal fin rays 5 to 7 from the west coast fish were excised to include the entire ray structure (including distal pterygiophores) or as close to the base as possible. Fin rays with prior damage exhibit scar-like markings at the point of damage. None of the samples used in this study exhibited such markings. The excised rays were placed in labelled plastic bags on ice and ultimately stored in a freezer prior to processing. Whenever possible, total length (TL), total weight and age estimates based on otoliths were determined for these individuals.

East coast fish were captured using hook and line in collaboration with an on-going study to determine their age structure throughout Florida using non-lethal techniques (Koenig et al., 2011). Once onboard, the total length was measured and the fish was doubly tagged with uniquely-numbered external (live-stock) and internal (Passive Integrated Transponder) tags. Dorsal fin-rays 5 to 7 were collected and processed as described above. Stomach contents were also collected non-lethally by manually removing partially digested prey items.

Sample processing

Fin rays were thawed in a drying oven for 4 h at a temperature of 55 °C. Once the samples had thawed, fatty tissue was removed using forceps. Each fin ray was then soaked in 30% hydrogen peroxide (H2O2) for 5 min to loosen the soft tissue surrounding the rays. Skin and membranes were cleaned from the rays using forceps and paper towels. The cleaned rays were glued to a petrographic microscope slide using Crystalbond (SPI Supplies, West Chester, Pennsylvannia, USA). A set of two cross sections (1.5 mm thick) were cut from the fin rays using a Buehler IsoMet slow-speed saw (Buehler, Lake Bluff, Illinois, USA). The purpose of these cross sections was to isolate individual annuli for stable isotope analysis. These cross sections were then sliced perpendicular to the first cut to create rectangular bands that represented the time series of the entire life of the fish (Fig. 2A). The slices were cut using a modified feather-blade guillotine. By inserting a spacer and a second parallel blade, the rectangular slices were cut from the initial cross sections of the fin ray. The rectangular slice was then cut using the single blade of the feather-blade guillotine to mechanically separate the rectangular slice into smaller pieces, each of which comprised a single annulus (or two annuli if the sample was too small to separate individual annuli, Fig. 2B). When the smaller pieces comprised two annuli the mean of the two ages was used, and the associated values were presented as such.

Figure 2 Mechanical separation of annuli.

Separation of annuli from a cross section of a dorsal fin ray of Atlantic Goliath Grouper, Epinephelus itajara. Dashed lines in (A) show where a rectangular section is taken from the cross section. Solid lines in (B) indicate excision lines separating individual annuli. Scale bars represent 1 mm.

Annuli were analyzed for bulk molar concentrations of carbon and nitrogen (C, N, C:N), and stable isotope abundance values, calculated as defined below (δ13C, δ15N). A 200 to 1,200 µg sample of each cross section was weighed on a Mettler-Toledo precision micro-balance (Mettler-Toledgo, Columbus, Ohio, USA), encapsulated in tin and loaded into a Costech Technologies Zero-Blank Autosampler (Costech Technologies, Montréal, Quebec, Canada). Samples were combusted at 1,050 °C in a Carlo-Erba NA2500 Series-II Elemental Analyzer (EA) (Thermo Fisher Scientific, Waltham, Massachusetts, USA) coupled in continuous-flow mode to a Finnigan Delta Plus XL isotope ratio mass spectrometer (IRMS) (Thermo Fisher Scientific, Waltham, Massachusetts, USA) at the University of South Florida, College of Marine Science. Stable isotopic compositions were expressed in per mil (‰) using delta notation: e.g., δ15N = (Rsample/Rstandard)-1]; where R=15N/14N. We calibrated the C:N measurements and δ13C and δ15N were normalized to the AT-Air and VPDB scales, respectively, using NIST 8573 (USGS 40; δ15N = − 4.52‰ ± 0.12‰; δ13C = − 26.39‰ ± 0.09‰) and NIST 8574 (USGS 41; δ15N = 47.57‰ ± 0.22‰; δ13C = 37.63‰ ± 0.10‰) L-glutamic acid Standard Reference Materials. All reference materials were sourced from the National Institute of Standards and Technology, U.S.A. Analytical precision, estimated by replicate measurements of a laboratory working standard (NIST 1577b Bovine Liver SRM, N = 31; δ15N = 7.83‰ ± 0.16‰; δ13C = − 21.69‰ ± 0.14‰), was ±0.13 δ13C, 0.18‰ δ15N, and ±0.25 C:N.

De-mineralization of fin rays

In an attempt to eliminate the carbon noise associated with the inorganic matrix (due to unpredictable substitutions between carbonate and phosphate), we tested whether de-mineralization of the fin rays was a feasible preparation technique to obtain values for both δ13C and δ15N that only measured concentrations in the organic matrix. While demineralization is often performed to isolate an organic matrix, recent studies suggest that the chemical process may alter the organic components of a sample, specifically δ13C and δ15N values (Rude, Smith & Whitledge, 2014). Fin rays (n = 21) were initially cleaned as described above, split into two halves and then sectioned at a 1.5 mm thickness. One of the two sections from each fin ray was then chosen at random for the de-mineralization process. Samples that were demineralized were sonicated in “ultra-pure,” milli-Q (Millipore, Billerica, Massachusetts, USA) water for 5 min and then submerged in 2% HCl for 24 h. After 24 h, the HCl was replaced and the samples were soaked for an additional 24 h. Samples were then rinsed thoroughly with distilled water, dried and sectioned (1.5 mm thick). All cross sections from both de-mineralized and control samples were powdered using a mortar and pestle to ensure uniformity within each sample. Samples were then weighed, encapsulated, and run on the EA-IRMS, as described above.

Data analysis

All statistical analyses were performed using MATLAB version R2012b. Age and size distributions between the two sample sets (west and east coasts) were plotted and analyzed using a two sample Kolmogorov–Smirnov test. Paired de-mineralized and control samples were analyzed for differences in two ways. A paired, two-tailed t-test was used to test overall differences of δ13C and δ15N values between the de-mineralized and control data sets. A procrustes analysis was used as an orthogonal least-squares analysis between the de-mineralized and control data sets by minimizing the sum of squares between the two. The symmetric orthogonal procrustean statistic (m2) was calculated as a goodness of fit between the control data set, and the de-mineralized data set. Values of m2 can range between 0 and 1, with lower values indicating a better fit between two sets of data. The procrustes analysis was able to test differences between each individual paired-sample.

In order to test whether the isotopic ratios were conserved over time, a two-tailed t-test was conducted using the δ15N values of the annuli corresponding to age 4 for two age groups of individuals within each study location. Here, we focused on age 4 to maximize sample size (n = 27), by choosing an annulus that was most commonly represented among samples (27 out of a possible 30). Samples within coastal groups were split into “young” (≤8 years old) and “old” fish (>8 years old) based on age at time of capture. The two groups were analyzed for differences in δ15N values to test whether the fin rays of older fish displayed signs of isotopic change over time. If δ15N values degraded over time in fin rays, then we would expect to see differences between the two groups as the signals in older fish would have had more time to change.

The chronologies of δ15N were created for each of the 30 individuals and grouped by coastal origin. Isotopic values were plotted against age (as determined by annuli) to investigate whether life-history shifts were indicated by changes in the δ15N values of individuals over time. Isotopic shifts were theorized to be most evident when the fish moved out of their nursery habitat at roughly 5 to 7 years of age (Koenig et al., 2007), due to either dietary shifts, shifts in background δ15N levels, or a combination of both. Given the repeated measures aspect of these data, a non-linear mixed-effects model based on the logistic equation was generated to model the distribution of data between δ15N values and age. The model predicted three parameters (response coefficient, y-intercept and horizontal asymptote) for each fish. These values were averaged to produce parameters for each population. F-ratios were calculated to compare between sample sets as a measure of goodness-of-fit. A permutation based p-value was calculated based on the F-ratio. In addition to the model comparison over the entire lives of each individual, a paired t-test was used to compare the “nursery habitat” life stage (≤6 years) and the “adult habitat” life stage (>6 years) to test ontogenetic shifts during a presumed migration period. Last, δ15N values were plotted against total lengths and age at time of capture for all fish and linear least squares regressions were calculated for both comparisons. Individual δ15 N values at time of capture were compared to total length of each specimen via a linear least squares regression of TL with the outer-most annulus to test for differences in adult feeding patterns between coasts. Correlations were then compared between the two sampling regions to test whether δ15N values consistently changed with size or age among all individuals.

Results

Total lengths of west coast fish ranged from 62 to 205 cm, that of east coast fish from 122 to 211 cm (Table 1). West coast samples were 2 to 18 years old and east coast samples ranged from 6 to 19 years old (Fig. 3). A two-sample Kolmogorov–Smirnov test verified that age structure did not differ between the two sample sets (ks = 0.3, p = 0.43), although size structure did due to several smaller individuals among the west coast samples (ks = 0.56, p = 0.01). None of the individuals analyzed were recaptures from previous sampling efforts.

Figure 3 Sample age distributions.

Percent frequency distributions of ages of Atlantic Goliath Grouper, Epinephelus itajara, sampled from the east coast (gray, n = 17) and west coast (black, n = 13) of Florida.

Table 1 Goliath grouper samples.

Age, length and location of all samples of Atlantic Goliath Grouper, Epinephelus itajara, included in the study.

Sample location	Total length (cm)	Age (years)	
West coast	62	2	
West coast	83	4	
East coast	122	6	
West coast	112	6	
East coast	174	8	
East coast	146	8	
East coast	162	8	
West coast	120	8	
West coast	126	8	
West coast	145	8	
West coast	124	8	
East coast	157	9	
East coast	147	9	
East coast	162	9	
East coast	168	9	
East coast	178	9	
West coast	130	9	
West coast	151	9	
East coast	171	10	
East coast	180	11	
East coast	185	12	
East coast	182	12	
East coast	195	14	
East coast	200	14	
West coast	198	14	
West coast	190	16	
West coast	190	16	
East coast	197	18	
West coast	205	18	
East coast	211	19	

De-mineralization

De-mineralization introduced strong and non-systematic artifacts, with samples having inconsistent loss of the light or heavy isotope for both δ13C (t = 2.02, df = 20 p = 0.009) and δ15N (t = 2.02, df = 20, p = 0.004). The procrustes analysis (Fig. 4) further supported that alterations during the de-mineralization process were variable to both δ13C and δ15N values (m2 = 0.64, p = 0.001). Differences in isotopic values between paired samples ranged from ±0.01 to ±3.14 for δ13C values and from ±0.05 to ±1.19 for δ15N values. The variable effects of demineralization to both δ13C and δ15N values precluded the use of a correction factor for treated samples. Due to these differences, demineralization was deemed inappropriate for this study, and carbon values were dismissed.

Figure 4 Effects of de-mineralization.

Procrustean superimposition plot of de-mineralized and control paired samples of fin rays from Atlantic Goliath Grouper, Epinephelus itajara. The dimensions describe the rotated data of the δ13C and δ15N values before and after transformation. The “scaled X” points are the initial values of the rotated data, of each non-demineralized sample. Residual lengths indicate the amount of difference between the samples.

Isotopic conservation and δ15N values at age

The δ15N values at the annulus corresponding to age 4 did not differ between young and old fish for either west coast (t = 2.26, df = 9, p = 0.46) or east coast samples (t = 2.14, df = 14, p = 0.20; Fig. 5). An ad hoc power analysis demonstrated high power for both west (power = 0.99) and east (power = 0.96) coast samples.

Figure 5 Degradation test for “young” and “old” fish.

Mean (se) δ15N values for the annulus corresponding to age 4 in both young and old Atlantic Goliath Grouper, Epinephelus itajara, from east and west coasts of Florida. Note differences in mean δ15N values from fish sampled from the east and west coasts of Florida.

The values of δ15N for west coast samples varied from 8.39 to 12.61‰(range = 4.22) and from 9.71 to 14.41‰(range = 4.70) for east coast samples. However, one outlier (>3 SD from the mean) was responsible for the larger range associated with east coast samples. Once the outlier was removed, the values ranged from 12.16 to 14.41‰(range = 2.25), roughly half that of west coast samples.

The δ15N values of both sample sets increased as the fish aged (Fig. 6). A general increase was observed during the presumed nursery life stage (i.e., at approximately ages 0–7 years) which then leveled off once the fish moved into their adult-habitat life stage. A paired t-test confirmed higher values for adult compared to juvenile stages (t = 2.16, df = 12, p < 0.001 for west coast samples and t = 2.12, df = 16, p = 0.004 for east coast samples). However, the non-linear mixed effects model highlighted differences between the two populations with regard to the response coefficients and horizontal asymptotes (F = 6.34 × 103, p = 0.001, coef = 2.35, asymptote = 11.98 for west coast samples and F = 5.57 × 104, p = 0.001, coef = 1.44, asymptote = 13.33 for east coast samples, Fig. 7).

Figure 6 Isotope chronologies for each individual.

δ15N values for all sampled annuli (one annulus = one data point) excised from dorsal fin rays of all sampled Atlantic Goliath Grouper (Epinephelus itajara) from both east (n = 17; open triangles) and west (n = 13; filled circles) coasts of Florida. Each line represents annuli from a single individual. When the separation of individual annuli was not possible, the average of the two was presented.

Figure 7 Average isotope chronologies by coast.

Mean (se) values of δ15N at age for west (filled circles) and east coast (open triangles) Atlantic Goliath Grouper, Epinephelus itajara. The single outlier (open squares) from the east coast is presented separately from the mean values for east coast fish. Trend lines represent the average non-linear mixed effects model for each population.

δ15N values at time of capture

δ15N values were positively related to total length for both west (coef(se) = 0.07(0.03); t = 2.3, r2 = 0.33, p = 0.05) and east coast fish (coef(se) = 0.15(0.02); t = 6.5, r2 = 0.75, p = 0.001; Fig. 8A). δ15N values were positively related to age for east coast fish (coef(se) = 2.93(1.16); t = 2.53, r2 = 0.31, p = 0.02), but not for west coast fish (coef(se) = 0.05(0.03); t = 1.73, r2 = 0.21, p = 0.14; Fig. 8B).

Figure 8 Nitrogen values at time of capture.

δ15N values of the outer-most annulus for both sample sets of Atlantic Goliath Grouper, Epinephelus itajara. Isotopic values were plotted against length (A) and age (B) at time of capture. δ15N was positively related to total length on both coasts, and age for east coast fish.

Discussion

Isotope chronologies

Fin-ray analysis is a non-lethal methodology that can be used to track isotopic chronologies in fishes. Organic isotopic-chronologies have previously been explored via annuli chronology in living tissues, such as trees and corals (McCarroll & Pawellek, 2001; McCarroll & Loader, 2004; Risk et al., 2009; Andreu-Hayles et al., 2011). To our knowledge, the current study represents the first to use a non-lethal method to derive an organic isotope chronology from a calcified tissue in fishes, without aging biases. The shift in isotopic values over time generated in the current study coincide with presumed life history events for Atlantic Goliath Grouper and may indicate that little to no tissue-turnover occurred as annuli were deposited in fin rays.

The use of isotopic analysis on fin rays required several assumptions, most notably, that fin-ray annuli corresponded with the age of the fish. This correspondence has been demonstrated for Atlantic Goliath Grouper (Brusher & Schull, 2009; Murie et al., 2009; Koenig et al., 2011) and other fishes (McFarlane & King, 2001; Sun, Wang & Yeh, 2002; Debicella, 2005; Murie & Parkyn, 2005; Muir et al., 2008; Khan & Khan, 2009; Glass, Corkum & Mandrak, 2011). Second, we assumed that the measured δ15N corresponded directly to the age associated with each annulus within individuals. Similar to the assumptions made in otolith micro-chemical analysis (Campana, 1999), we assumed that the chemical constituents of each annulus were representative of the fish’s diet at that age.

The limitations of the technique we used were classified as either mechanical or isotopic. The mechanical limitations were largely due to the size and composition of cross sections of fin rays. Curved annuli were cut with a straight blade, which made separating annuli difficult. Even with the relatively large samples used in this project, roughly ten samples had annuli that were coupled. No samples were excluded from the analysis due to this constraint. However smaller fishes with narrower annuli than Atlantic Goliath Grouper would present a new challenge and would require instrumentation with higher precision. In addition, two main isotopic limitations were apparent in the current study. The absence of δ13C values from the study represented an important constraint, given its relevance to understanding basal resources. The carbonated hydroxyapatite that makes up the inorganic matrix of fin rays (Mahamid et al., 2008; Mahamid et al., 2010) introduced carbonate analytes into control samples. These carbonates replaced the phosphate molecules that make up the structural component of fin rays. This carbonate replacement is in non-stoichiometric equilibrium, and cannot be quantified consistently. Intact fin rays will thus have inorganic-carbon components that are variable among samples. De-mineralization was deemed inappropriate for the current study, based on the altered and inconsistent offsets of both δ13C and δ15N values as highlighted in the procrustes analysis. Consequently, δ13C values were excluded from the analysis. Future studies may choose to test the de-mineralization process further by powdering samples prior to acid treatments. Powdering the sample prior to de-mineralization may facilitate a more complete digestion of all the inorganic carbon in the sample. The second isotopic limitation we faced dealt with the source of the observed δ15N shifts over time. The δ15N chronologies in this study match the life-history characteristics of Atlantic Goliath Grouper observed in previous studies, although the magnitude of the shifts differed between coasts. The difference in δ15N values between the juvenile and adult stages was consistent with the ontogenetic movement patterns and dietary shifts previously documented (Eklund & Schull, 2001; Koenig et al., 2007). The composition of prey differ taxonomically between juveniles (mainly crabs) and adults (mix of crabs, lobsters, fishes, mollusks and echinoderms), largely due to species composition at the different habitats (Koenig & Coleman, 2010). The observed isotopic variations may have been due to an altered diet at sequential ontogenetic events, to a shift in isotopic baseline values at different locations or a combination of both. Isotopic background values differ between locations, and can thus influence measured isotopic values if a fish moves from one area to another (McMahon, Hamady & Thorrold, 2013; Radabaugh, Hollander & Peebles, 2013). Further study using compound-specific analysis of amino acids could potentially differentiate between δ15N variation due to diet or to background isotopic differences (McClelland & Montoya, 2002; Chikaraishi et al., 2007; Loick, Gehre & Voss, 2007; Ellis, 2012).

Population differences

The relationship between δ15N values and age differed between the two sampling regions. Similar-aged Atlantic Goliath Groupers exhibited 0.5 to 6.0 δ15N value enrichment on the east coast compared to the west coast. Without an isoscape effect (e.g., Radabaugh, Hollander & Peebles, 2013), east coast Atlantic Goliath Groupers would be expected to feed at 1 to 2 trophic levels higher than west coast fish, which is unlikely. To our knowledge, such a drastic difference in trophic level has never been documented between local populations of a conspecific, nor has concurrent research found differences between the stomach contents of individuals between the two coasts (Koenig et al., 2011). The clear division in overall δ15N values can most likely be attributed to isoscape effects, meaning that isotopic background levels change among locations depending on ambient conditions (Graham et al., 2010). The data presented here suggest that an isoscape effect may be detectable between the two coasts of Florida. Isoscape effects naturally occur over large spatial scales, such as open ocean environments (Graham et al., 2010), but river outflows can influence isotopic baselines of ocean basins (Radabaugh, Hollander & Peebles, 2013) while near-shore environments can be directly influenced by anthropogenic activities (Seitzinger et al., 2005). Elevated δ15N values in the east coast fish may be partially due to the higher abundance and density of the human population than on the west coast of Florida (U.S. Census Bureau, 2010). Anthropogenic inputs, such as treated sewage released into east coast waters has been documented to elevate background δ15N values, particularly near centers of high human population density (Lapointe et al., 2005a; Lapointe et al., 2005b; Risk et al., 2009).

Both west and east coast fish exhibited a shift in isotopic values from juvenile to adult life stages, but west coast individuals did so faster and with a lower asymptote. The elevated δ15N signals from the isoscape effect may once again contribute to the observed patterns. One pragmatic explanation may be that both populations exhibit an isotopic shift over time, but the patterns observed in east coast samples were swamped by isoscape effects. Indeed, the δ15N values of several individuals in their juvenile phases are similar to that of their adult values. Future studies should aim for a community-level assessment to test whether lower values exist throughout the food web on the west coast of Florida in comparison to the east coast. High baseline δ15N values would enrich all the organisms within a food web, particularly those close to shore, where the juvenile Atlantic Goliath Grouper reside. The individual outlier on the east coast may have migrated from a west coast nursery, as rare instances (<5%) have been observed for individuals making migrations of those distances (Koenig, Coleman & Kingon, 2011). In contrast, a larger area of mangrove habitat, with a potentially different isoscape, exists on the west coast (Fig. 1). Extensive mangrove habitat has been documented as the primary nursery habitat for Atlantic Goliath Grouper (Eklund & Schull, 2001; Koenig et al., 2007; Gerhardinger et al., 2009; Lara et al., 2009). The southwest coast of Florida is dominated by mangrove habitat that has been suggested as the most densely populated nursery habitat for Atlantic Goliath Grouper in their range (Koenig et al., 2007; Koenig, Coleman & Kingon, 2011).

The δ15N values associated with total length may have also been affected by differences in ecosystem dynamics between the two coastal shelves. Previous studies have documented advanced sexual maturity at shorter total lengths of several fish species in the eastern Gulf of Mexico compared to other nearby regions (Gartner, 1993). If this trend holds true for Atlantic Goliath Groupers, then we would not expect to observe a significant relationship between total length and δ15N values on the west coast, because development within individuals would be variable. The mechanism for this phenomenon of unpredictable development in the region is not clearly understood, and warrants further investigation.

Implications and significance

The chemical analysis of fin rays represents a powerful method to better understand life-history movements and trophic shifts of fishes. Moreover, this approach is non-lethal and uses fewer samples than other, common techniques. These benefits help to promote a methodology that can facilitate important studies on endangered fishes around the world, as exemplified here with the Atlantic Goliath Grouper. Although a controlled laboratory experiment would be ideal to test isotopic chronologies in fin rays of teleosts, such an experiment is not practical for longer lived fishes. One viable alternative would be to test fin rays of fishes that have been in captivity for an extended period of time. A direct comparison could be made between life history stages that occurred in the wild versus those that occurred while in captivity. Unless the isotopic background levels as well as the isotopic values of the food source were exactly the same, then these two life history stages should differ within individual fishes. Our efforts here represent an observational basis to justify such a study. Studies that examine life history processes via non-lethal sampling can be used in turn to influence management strategies by gaining a better understanding of age-specific characteristics of species of interest. Changes in such characteristics over time are the result of altered movements, diet or both. This knowledge can be used in turn to direct both habitat and community-level conservation practices.

Supplemental Information

Supplemental Information 1 Raw data

The raw data that was used in this study is presented in this excel file.

Click here for additional data file.

East coast samples were collected as approved by permits granted by the Florida Fish and Wildlife Commission (SAL-12-1244A-SRP) and the National Oceanic and Atmospheric Administration (F/SER24:PH). West coast sample collection was greatly facilitated by A Collins of FWRI. We thank E Peebles, D Jones, B Walther, B Ruttenberg and S Murawski for conceptual advice; P Hallock-Muler, J Curtis and T Pusack for comments; and D Jones, M Drexler, M Albins and J Kilborn for analytical advice.

Additional Information and Declarations

Competing Interests

Author Contributions

Animal Ethics

Field Study Permissions

The authors declare there are no competing interests.

Orian E. Tzadik conceived and designed the experiments, performed the experiments, analyzed the data, contributed reagents/materials/analysis tools, wrote the paper, prepared figures and/or tables, reviewed drafts of the paper.

Ethan A. Goddard performed the experiments, reviewed drafts of the paper.

David J. Hollander contributed reagents/materials/analysis tools, reviewed drafts of the paper.

Christopher C. Koenig contributed reagents/materials/analysis tools, reviewed drafts of the paper, guidance on methodology.

Christopher D. Stallings conceived and designed the experiments, reviewed drafts of the paper.

The following information was supplied relating to ethical approvals (i.e., approving body and any reference numbers):

1. Institutional Animal Care and Use Committee (IACUC), Division of Research Integrity and Compliance, University of South Florida.

2. 4193W.

The following information was supplied relating to field study approvals (i.e., approving body and any reference numbers):

1. Florida Fish and Wildlife Commission; National Oceanic and Atmospheric Administration.

2. SAL-13-1244A-SRP; F/SER24:PH.

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
