# Peer review of "Non-lethal approach identifies variability of δ15N values in the fin rays of Atlantic Goliath Grouper, Epinephelus itajara"

_PeerJ, doi:10.7717/peerj.1010_

## Round 0.1 · original submission · Minor Revisions

Please consider all the suggestions made by the reviewers in the revised version of your manuscript.

Reviewer 1 ·

Basic reporting

The authors present a well thought out and competently executed piece of work. But for some minor revision this manuscript could be accepted for publication as is. To this end the authors' attention is drawn to the below comments.
Throughout the manuscript, formatting style and stable isotope nomenclature needs to be comply with IUPAC guidelines and recommendations. For example, the delta symbol must be presented as a Greek symbol formatted in Italics. Also, if reporting delta-values in ‰ (see above) there must be a space separating the number from the ‰ sign (see below).
Page 7, line 173: Given the definition of the delta-value (see page 7, line 179) it should be clear it is not a stable isotope ratio but a stable isotope abundance value calculate from a ratio of ratios.
Page 7, line 179: Authors (and editors) should note the latest IUPAC guidelines and recommended terms of stable isotope ratio measurements and reporting results thereof now give the equation defining the delta notation without the extraneous factor 1,000 (Coplen, T.B., 2011, Rapid Commun. Mass Spectrom., 25, 2538; Brand, W.A. et al., 2014, Pure Appl. Chem., 86, 425). Reporting delta-values as ‰ values merely means a figure of -0.02592 is expressed as 25.92 x 10exp-3 or 25.92 ‰ with a space between number and ‰ sign.
Page 7, line 181: NIST SRMs 8573 and 8574 are commonly known as USGS 40 and USGS 41 and should be referred to as such. In addition, it is good practice to report (in brackets) their respective delta13C and delta15N values together with information where these reference materials were sourced from.
Page 7, line 183: In addition to the information on measurement precision provided on the basis of replicate analyses of NIST SRM 1577b it would be useful if the authors would state their in-house consensus delta13C and delta15N values for this material.

Experimental design

Stable isotope analytical procedures as described by the authors meet with requirements for resulting data to be traceable and thus internationally comparable. This reviewer is particularly happy with the authors' use of 2 scale anchors and 1 quality control.
From the description given in the experimental section it would seem fin samples were powdered after the attempts to demineralize them. This would of course explain the spurious delta13C and delta15N values the authors observed for demineralized samples. Exposing powdered fin material to dilute HCl (or better HOAc as is the practice for e.g. archaeological tooth samples) might have yielded more consistent results.

Validity of the findings

Was the data point for the East Coast outlier included in the data analysis presented here? Figures 8A and 8B both seem to include this data point (~120+ cm/age 6 v. delta15N value of ~10.3 ‰). If so, why was this individual not excluded from the data analysis given it had already been classified as an outlier?

Additional comments

Would the authors care to speculate about potential reasons for the East Coast outlier? One or two probable scenarios spring to mind given the close proximity of its delta15N values to age corresponding delta15N values of West Coast fish.

Annotated reviews are not available for download in order to protect the identity of reviewers who chose to remain anonymous.

Reviewer 2 ·

Basic reporting

This work sets out to provide and test a novel non-lethal stable isotope technique for large, long-lived, and threatened reef fish, with an emphasis on the Atlantic Goliath Grouper off the coasts of Florida. The authors stress the usefulness of this technique to provide information essential to the management of any fish species of concern. As the title claims, this work demonstrates variability of nitrogen values in the fin rays of the Atlantic Goliath Grouper; however, the work provides a limited explanation to the importance and use of such data to fisheries management.

The authors highlight that this study represents the first to use a non-lethal method to derive an organic isotope chronology from a calcified tissue in fishes, without aging biases. Although this warrants merit, due to its inability to accurately measure carbon isotope levels and account for variation of n nitrogen base levels between sites, further optimization is required before this method becomes a standard in fisheries research.

This work is suitable for publication, but needs to place more emphasis on the limitations of the technique and how to overcome them prior to claiming success. Furthermore, it is unclear what the major research question being addressed is.



FIGURES
Fig. 2.
Both images should have a scale bar.
Fig. 3.
According to your methods, 17 fish were sampled from the east coast and 13 were sampled from the west coast. This is in disagreement with your caption and needs to be fixed.
Fig. 4.
This is a sloppy figure and hard to interpret. What is meant by scaled X (is x both carbon and nitrogen values)? Can you use different symbols or colors to distinguish between carbon and nitrogen samples?
Fig. 6.
It appears that not every annuli for a single fish was ever analyzed. Was this due to mechanical issues and if so why is this not mentioned in the methods? The methods section gives the impression that all or nearly all annuli were analyzed from a single individual. Furthermore, some east coast fish appear to have very high nitrogen levels corresponding to young ages, values that are as high as or higher than their 8-9 year values. What might account for this?
Fig. 8
Is the single outlier from the east coast included in the trend lines for both total length and age?

Experimental design

MATERIAL AND METHODS
Study Area
Since all of the West coast fish were from opportunistic “fish-kills” why weren’t eye lenses also used?
If there is a readily-available amount of material from opportunistic “fish-kills”, and East Coast and West Coast fish exhibit similar patterns of increased nitrogen signatures with age, why is there a need for developing a non-lethal sampling method? How common are opportunistic “fish-kills” on the Florida coast?

Tissue Selection and Sample Collection
Why dorsal fin rays over other fin rays? Why dorsal fin rays 5-7?
Is there a way to determine if fin rays had previously been damaged and then regrown? Might this influence your results?

Sample Processing
How many samples consisted of two annuli and were they treated the same as a single annulus in your analyses?

Validity of the findings

DISCUSSION
Isotopic Chronologies
In mentioning challenges in separating annuli, it’s unclear if this may have affected the results presented here. How many samples were used that contained two or more annuli and/or how many were excluded from your analyses for the same reason?
The inability to use carbon isotopes with fin rays is a major concern of this methodology and little is mentioned on how this can be overcome in future studies. Should this method then only be considered for investigating nitrogen isotopic values and if so, does this limit the usefulness of this method as well?
The inability to explain differences in baseline nitrogen levels is also of concern, especially where increased baseline levels are a result of anthropogenic effects. Taking a more community level approach and collecting samples of known prey items and other trophic guilds (autotrophs, herbivores, predators, and detrivores) from the same collection sites could provide context for nitrogen levels. Was there a difference between the stomach contents of the non-lethally collected East coast fish and the “fish-kill” specimens from the West coast?
It remains unclear how this study/proposed methodology adds to existing knowledge of life histories of the Atlantic Goliath Grouper (the trend in increased nitrogen levels corroborates with what was previously known of ontogenetic transitions from estuarine to reef habitats), and how this could influence management strategies of the fishery. I feel this should be a major focus of this paper, and more emphasis should be given to this subject.

---

## Round 0.2 · accepted · Accept

The authors have improved the manuscript, I am happy to accept it